# Filarial DAF-12 sense the host serum to resume iL3 development during infection

Rémy Bétous[1]*, Anthony Emile[1], Hua Che[2], Eva Guchen[1], Didier Concordet[1], Thavy Long[2], Sandra Noack[3], Paul M. Selzer[3], Roger Prichard[2], Anne Lespine[1]*

1 INTHERES, Université de Toulouse, INRAE, ENVT, Toulouse, France, 2 Institute of Parasitology, McGill University, Sainte-Anne-De-Bellevue, Canada, 3 Boehringer Ingelheim Animal Health, Ingelheim am Rhein, Germany

* remy.betous@inrae.fr (RB); anne.lespine@inrae.fr (AL)

**Data Availability Statement:** All relevant data are within the manuscript and its Supporting information files.

**Funding:** This study was supported by the department of Animal Health at INRAE (newly

## Abstract

Nematode parasites enter their definitive host at the developmentally arrested infectious larval stage (iL3), and the ligand-dependent nuclear receptor DAF-12 contributes to trigger their development to adulthood. Here, we characterized DAF-12 from the filarial nematodes *Brugia malayi* and *Dirofilaria immitis* and compared them with DAF-12 from the non-filarial nematodes *Haemonchus contortus* and *Caenorhabditis elegans*. Interestingly, *Dim* and *Bma*DAF-12 exhibit high sequence identity and share a striking higher sensitivity than *Hco* and *Cel*DAF-12 to the natural ligands Δ4- and Δ7-dafachronic acids (DA). Moreover, sera from different mammalian species activated specifically *Dim* and *Bma*DAF-12 while the hormone-depleted sera failed to activate the filarial DAF-12. Accordingly, hormone-depleted serum delayed the commencement of development of *D. immitis* iL3 *in vitro*. Consistent with these observations, we show that spiking mouse charcoal stripped-serum with Δ4-DA at the concentration measured in normal mouse serum restores its capacity to activate *Dim*DAF-12. This indicates that DA present in mammalian serum participate in filarial DAF-12 activation. Finally, analysis of publicly available RNA sequencing data from *B. malayi* showed that, at the time of infection, putative gene homologs of the DA synthesis pathways are coincidently downregulated. Altogether, our data suggest that filarial DAF-12 have evolved to specifically sense and survive in a host environment, which provides favorable conditions to quickly resume larval development. This work sheds new light on the regulation of filarial nematodes development while entering their definitive mammalian host and may open the route to novel therapies to treat filarial infections.

## Author summary

Nematode parasites infect their definitive mammalian hosts at a developmentally arrested infectious larvae stage (iL3). Upon infection, development resumption is controlled by the ligand-dependent nuclear receptor DAF-12. It has been shown, in several nematode species, that host cues stimulate the synthesis, by the larvae, of the DAF-12 ligands dafachronic acids. Here, we investigated if this pathway is concerved in the phylum of filarial nematodes. We compared DAF-12 from different nematode species and we found that

recruited researchers grant) to (R.B.); and a grant
from the Natural Sciences and Engineering
Research Council of Canada (Grant No. RGPIN-
2017-04010) to (R.P.). The funders had no role in
study design, data collection and analysis, decision
to publish, or preparation of the manuscript.

**Competing interests:** The authors have declared
that no competing interests exist.

filarial DAF-12 are much more sensitive to their ligands than DAF-12 from non-filarial
species. Moreover, we observed that filarial DAF-12 were specifically activated by mammalian sera. Importantly, depletion of DAF-12 ligands from serum by charcoal stripping
abrogated DAF-12 activation and impaired significantly the initiation of development of
filarial nematodes *in vitro*. Remarkably, addition of Δ4-DA at the concentration measured
in normal mouse serum restores the capacity of charcoal stripped serum to activate *Dim*-DAF-12. Finally, analysis of RNAseq data suggest that, at the time of infection, genes
encoding putative dafachronic acid synthesis enzymes are weakly expressed. These observations indicate that filarial nematodes have evolved to sense the mammalian host environment directly through DAF-12 to resume their development, and may open a route to
novel therapies.

## Introduction

*Dirofilaria immitis* is a filarial nematode belonging to the clade III superfamily Filarioidea
causing canine heartworm disease. *D. immitis* is evolutionarily related to other filarial nematodes such as *Brugia malayi* and *Onchocerca volvulus*, respectively responsible for human parasitic lymphatic and cutaneous filariases. *D. immitis* naturally infects canids, including
domestic dogs, coyotes and wolves, but it has also zoonotic potential [1]. However, the parasite
does not fully mature in humans. Like other filarial nematodes, *D. immitis* has no bacteria-feeding rhabditiform free-living stage and exhibits a complex lifecycle involving multiple
developmental stages in two successive hosts, the definitive mammalian host and an arthropod
vector (Aedes, Anopheles or Culex mosquitoes) [2]. *D. immitis* initiates its early developing
larval stages in mosquitos which transfer the infective third-stage larvae (iL3) to the dog definitive host during a blood meal, providing a suitable environment for the later developing larval
stages and adults. Females are ovoviviparous and release sheathless microfilariae, which correspond to a pre-L1 stage. Microfilariae remain developmentally quiescent in the mammalian
blood stream for up to 2–3 years, until taken up by a mosquito during a blood meal [3], for
another parasitic cycle. *D. immitis* infections have been identified throughout the world in
tropical and temperate regions [4,5]. It is the most medically important parasitic infection of
domestic dogs in the United States of America with no state free of infections [6]. Heartworm
disease prevention is achieved by the administration of macrocyclic lactones (MLs), targeting
the third and fourth larval stages (L3, L4) of the parasite [7], with minimal associated morbidity of the host. However, *D. immitis* ML-resistant isolates have emerged, especially in the
Lower Mississippi region, USA [8]. Therefore, novel drugs or approaches are urgently needed
to prevent *D. immitis* infections. To this end, studying the mechanisms that govern nematode
infection is an attractive means to reach this goal.

The nematode nuclear hormone receptor (NHR) DAF-12 represents an exciting potential
therapeutic target [9] as it controls strategic development stages, and represents therefore a relevant target to undermine parasitic pathogens. NHRs are transcription factors whose hallmark
is the presence of two conserved functional domains namely the DNA binding domain (DBD)
which binds to specific genes and the ligand binding domain (LBD) which can bind specific
hydrophobic ligands. The ligand-activated receptor undergoes conformational changes and
recruits transcriptional coregulators to modulate the transcription of sets of target genes.
Therefore, pharmacological inhibition of a single NHR can alter simultaneously the expression
of a large number of genes, compromising vital physiological functions. In this regards, as
DAF-12 controls the life cycle of parasites, its inhibition could interrupt the growth and the

reproduction of the worms. In the free-living nematode *C. elegans*, DAF-12 controls the decision to enter dauer diapause which is a quiescent alternative L3 stage (L3d), only established upon harsh conditions such as crowding, food deprivation or high temperature [10]. Entry and exit from the L3d is determined by the presence or absence of DAF-12 ligands, such as Δ4- and Δ7-dafachronic acids (Δ4- and Δ7-DA) [11]. In parasitic nematodes, iL3 are also able to survive under harsh conditions and are developmentally quiescent until they contact the host and resume their development, therefore the iL3 stage is thought to be equivalent to the L3d of free-living nematodes [12]. Importantly, DAs promotes iL3 development in different parasite species [13–15]. Besides, the administration of Δ7-DA stimulates *Strongyloides stercoralis* post-parasitic first-stage larvae (L1s) to develop to free-living adults instead of iL3 [13] and administration of Δ7-DA in combination with the ML, ivermectin, in the gerbil strongyloidiasis model resulted in a near cure of the infection [16]. However, whether these observations can be transposed to other nematode species remains unknown.

Recently, we identified *D. immitis* DAF-12 and showed that it exhibits a much higher affinity for DAs than any other nematode DAF-12 investigated so far. Importantly, *Dim*DAF-12 also responded to 3β-hydroxy-5-cholestenoic acid (CA) activation, a bile acid precursor found in human plasma, and exposing iL3 *D. immitis* larvae to DAs or CA stimulated their development toward the L4 stage [14]. Given the key role of *Dim*DAF-12 in development resumption of *D. immitis* iL3, we sought to determine if *Dim*DAF-12 characteristics and functions are shared with other filarial nematodes. We identified and cloned DAF-12 from the related filarial nematode *B. malayi* and found that it shares high sequence identity and strong ligand sensitivity with *D. immitis* DAF-12, compared with DAF-12 from non-filarial nematodes. Using a combination of *in vitro* activity reporter and development assays, we have shown that Δ4-DA, naturally present in mammalian sera, can directly and specifically activate filarial DAF-12 to trigger initiation of iL3 development. Lastly, transcriptomic data available on *B. malayi* suggest that the parasites may control the production of their own DAs along the development cycle, and switch off DA production at the infective iL3 stage. This work emphasizes the specificities of filarial parasite life cycle regulation and opens the route to a novel therapeutic strategy to prevent diseases caused by filarial nematodes.

## Results

### Analysis of filarial and non-filarial DAF-12 sequences

We recently identified and cloned the ortholog of *daf-12* from *D. immitis* [14], and we showed that the DBD of *Dim*DAF-12 is highly conserved. However, the LBD of *Dim*DAF-12 is more divergent with only 43% sequence identity to *Cel*DAF-12 LBD. The question arises as to whether the LBDs are evolutionary conserved within filarial nematodes. To address this question, we screened the *B. malayi* reference genome Bmal-4.0 (https://parasite.wormbase.org/Brugia_malayi_prjna10729) with DAF-12 from *D. immitis*, and we identified a *B. malayi daf-12* gene. We cloned its cDNA from iL3 tissues to obtain the true gene sequence and compare its predicted protein sequence with DAF-12 orthologs from *D. immitis*, *H. contortus* and *C. elegans* (Fig 1).

All DBD shared high identity (95–99%) (Fig 1A), suggesting that the proteins have closely related binding modes on the promoter regions of the target genes in the four nematode species. By contrast, the LBD sequences of DAF-12 are more divergent than the DBD. Alignment of the four LBD revealed conservation of only some portions corresponding to several α-helix (S1 Fig) as we observed previously when aligning *Dim*DAF-12 and *Sst*DAF-12 and *Ace*DAF-12 crystal structure sequences [14]. The LBD sequence of *Bma*DAF-12 which shares 95% amino acid sequence identity with *Dim*DAF-12 LBD, differs significantly from non-filarial

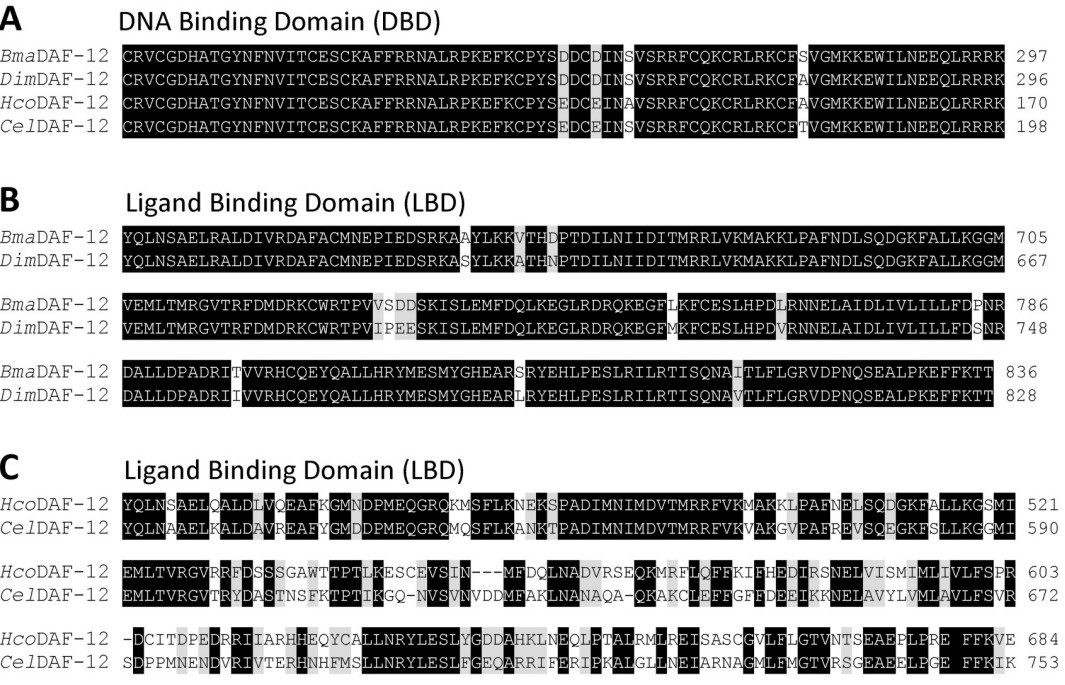

**Fig 1. Alignment of DAF-12 protein sequences from different nematode species.** (**A**) Sequence alignment of the DBD of *D. immitis* (*Dim*; accession no. MK820661), *B. malayi* (*Bma*; accession no. ON714136), *C. elegans* (*Cel*; NM_001029376), and *H. contortus* (*Hco*; MN017114). Identical residues are highlighted in black and similar residues are highlighted in grey. (**B**) **and** (**C**) Sequence alignment of the LBD. Multiple sequence alignment was performed with the MAFFT program from MyHits. Identical (black) and similar (grey) residues have been color coded with the Color Align Conservation program.

DAF-12 with respectively only 55% and 43% identity with *H. contortus* and *C. elegans* DAF-12 (Fig 1B and Table 1).

## Dafachronic and cholestenoic acids strongly and specifically activate filarial DAF-12

In order to study and compare the activity of DAF-12 from the different species, we used a transactivation assay based on mammalian cells co-transfected with the chimeric construct fusing the LBD of the receptor of interest (*Hco*, *Cel*, *Dim* or *Bma*DAF12) and the DBD of GAL4 along with a plasmid containing the luciferase reporter gene, which releases luminescence upon activation of the nuclear receptor by a natural or synthetic ligand. To validate the assay, we showed by Western-blot that all four constructs were expressed in transfected NIH-3T3 cells, at a level consistant with the transactivation assay efficiency (S2 Fig). Dose-response

**Table 1. Pairwise amino acid sequence comparison (% of identity) between DAF-12 LBD from *D. immitis* (*Dim*; MK820661), *B. malayi* (*Bma*; ON714136), *C. elegans* (*Cel*; NM_001029376), and *H. contortus* (*Hco*; MN017114).**

| | LBD identities (%) | | | |
|---|---|---|---|---|
| | *Dim*DAF-12 | *Bma*DAF-12 | *Cel*DAF-12 | *Hco*DAF-12 |
| *Dim*DAF-12 | - | - | - | - |
| *Bma*DAF-12 | 95 | - | - | - |
| *Cel*DAF-12 | 44 | 43 | - | - |
| *Hco*DAF-12 | 55 | 55 | 58 | - |

experiments confirmed that *Dim*DAF-12 exhibits much lower half maximal effective concentration ($EC_{50}$) for Δ7- and Δ4-DA than *Cel*DAF-12 (Fig 2A and 2D and Table 2).

Interestingly, we observed that Δ7- and Δ4-DA also activate *Bma*DAF-12 with high potency and their $EC_{50}$ were comparable with those of *Dim*DAF-12 (Fig 2B and Table 2). We confirmed that Δ7-DA activate *Hco*DAF-12 with same efficiency as *Cel*DAF-12 (Fig 2C and 2D and Table 2). Moreover, we showed for the first time that *Hco*DAF-12 can be activated by Δ4-DA (Fig 2C). Notably, while filarial DAF-12 have similar sensitivities to Δ7- and Δ4-DA, non-filarial DAF-12 exibit relatively higher sensitivity to Δ7-DA compared with Δ4-DA (Table 2), although still much lower than filarial DAF-12.

Using the cell-based transactivation assay, we previously demonstrated that CA [15] activates *Dim*DAF-12 with an $EC_{50}$ within the range of CA concentration measured in human plasma [14]. Interestingly, CA exhibited similar activation potency for *Dim* and *Bma*DAF-12 ($EC_{50}$ = 192 and 241 nM, respectively), whereas CA failed to activate *Hco* and *Cel*DAF-12 at concentrations well above the relevant physiological values (Fig 2 and Table 2). These results

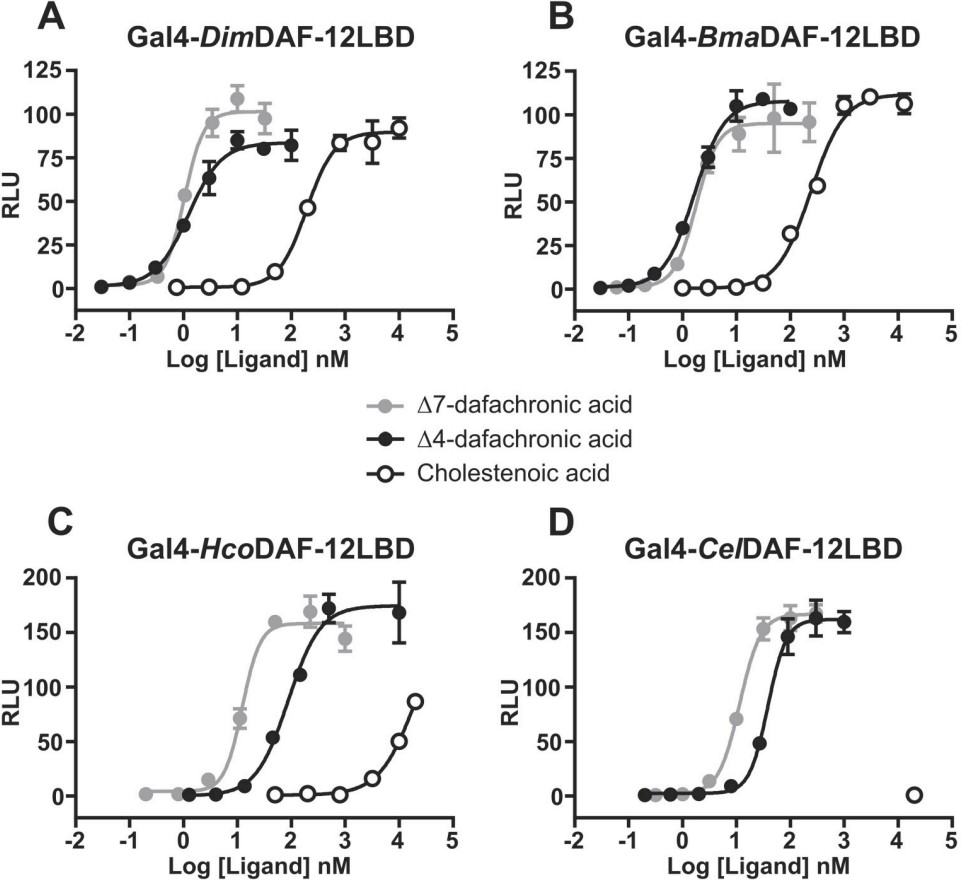

**Fig 2. Dose-response curves for Δ7-DA, Δ4-DA and CA on the activation of (A) *Dim*DAF-12, (B) *Bma*DAF-12, (C) *Hco*DAF-12 and (D) *Cel*DAF-12. NIH3T3 cells were co-transfected with Gal4-DAF-12_LBD and the luciferase gene reporter construct.** Recombinant cells were incubated with increasing concentrations of Δ4-DA, Δ7-DA or CA. The ligand binding and transactivation activity were assessed by measuring the luciferase activity, which was normalized by Renilla luciferase activity for transfection efficiency and expressed as relative light units (RLU). Data representing normalized luciferase activities are plotted with non linear regression fit using sigmoidal dose-response with variable slope (Prism 6.0, Graph Pad Software, Inc.). The figure shows one representative experiment out of two. Values are means of technical triplicates ± standard deviation.

**Table 2. EC$_{50}$ values of Δ4-DA, Δ7-DA and CA for *Dim*, *Bma*, *Hco* and *Cel*DAF-12.** EC$_{50}$ and the logEC$_{50}$ standard errors (SE) values were calculated from the nonlinear regression fits (sigmoidal dose-response with variable slope) of technical triplicates presented in Fig 2. nc: not calculated.

| | EC$_{50}$ (logEC$_{50}$ SE), *nM* | | | |
| --- | --- | --- | --- | --- |
| | *Dim*DAF-12 | *Bma*DAF-12 | *Cel*DAF-12 | *Hco*DAF-12 |
| Δ4-dafachronic acid | 1.2 (0.05) | 1.7 (0.03) | 38 (0.03) | 87 (0.06) |
| Δ7-dafachronic acid | 1.0 (0.03) | 1.8 (0.06) | 11 (0.02) | 12 (0.04) |
| Cholestenoic acid | 192 (0.03) | 241 (0.04) | nc | nc |

show that filarial DAF-12 are much more sensitive to their natural or exogenous ligands than *Hco* and *Cel*DAF-12.

## Filarial DAF-12 are activated by sera from different mammalian species

Charcoal stripping of serum is commonly used to selectively remove hormones that may otherwise interfere with the activity of the studied nuclear receptors. This is the reason why the cell-based transactivation assays are routinely conducted in standardized conditions using charcoal-stripped Fetal Bovine Serum (FBS) to assess accurately NHR ligand activities. To determine if filarial DAF-12 could be activated by ligands present in mammalian sera, we compared their activities in the presence of regular or charcoal-stripped FBS. DAF-12 from the two filarial nematodes, *D. immitis* and *B. malayi*, but not from *H. contortus* or *C. elegans* were activated by 10% of regular FBS, while charcoal stripped FBS did not activate DAF-12 from any of the nematode species (Fig 3A). Interestingly, both *Dim* and *Bma*DAF-12 responded to regular FBS in a dose dependent manner (Fig 3B) indicating that regular FBS contains authentic filarial DAF-12 ligands. As DAF-12 natural ligands are bile acid like molecules and because charcoal stripping efficiently removes steroid hormones (such as estradiol or progesterone) [17], these results suggest that filarial DAF-12 are sensitive to ligands contained in FBS with a cholesterol-derived core structure.

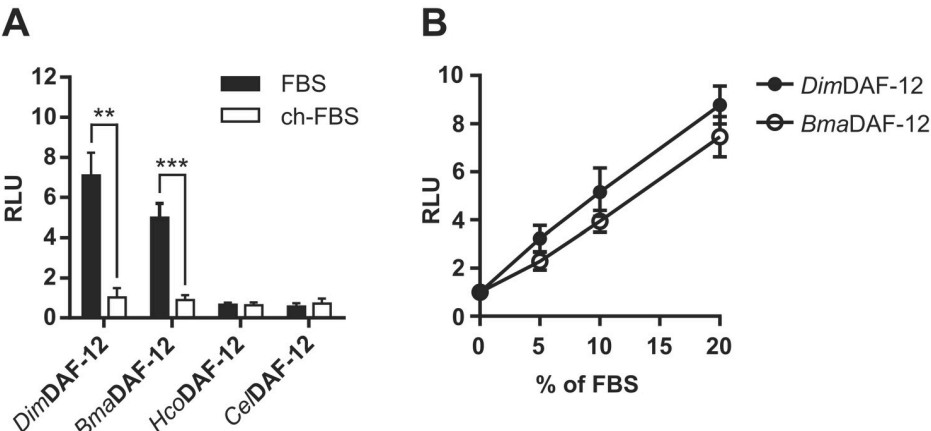

**Fig 3. FBS activates specifically *Dim* and *Bma*DAF-12.** NIH3T3 cells were co-transfected with Gal4-DAF-12_LBD from *Dim*, *Bma*, *Hco* or *Cel* and luciferase gene reporter construct and then **(A)** incubated with regular or charcoal-stripped FBS for 24 hours. **(B)** Effects of increasing amounts of FBS on activation of *Dim* and *Bma* DAF-12 activity. The values were normalized to the empty vector-transfected cells and expressed as relative light units (RLU). Data represent the average of normalized luciferase activity and the error bars correspond to the standard deviation of three independent experiments. The significance of the effects was analyzed by unpaired Student's t-tests using Prism 6.0 (Graph Pad Software, Inc.). **p < 0.01; ***p < 0.001.

To determine whether the capacity to activate filarial DAF-12 is specific to FBS, we prepared sera from different mammalian species including dogs and humans, which are respectively the favored vertebrate hosts of *D. immitis* and *B. malayi*, and we tested their ability to activate DAF-12. All mammalian sera tested were able to activate *Dim* and *Bma*DAF-12, but they failed to activate *Hco* and *Cel* DAF-12 (Fig 4A), indicating that all the mammalian sera contain ligands that activate specifically filarial DAF-12.

Notably, the activities of filarial DAF-12 were not equivalent for all the sera. For instance, pig sera activated significantly more filarial DAF-12 (2 to 10 times), than any other sera tested (Fig 4A) suggesting that either the nature or the amount of filarial DAF-12 ligands could be different according to the mammalian species. These differences were not the consequence of inter-individual variations since sera from different donors of the same species led to comparable activation of *Dim*DAF-12 (S3 Fig). Importantly, charcoal stripping of human, canine and porcine sera completely abolished the serum-mediated activation of *Dim*DAF-12 and

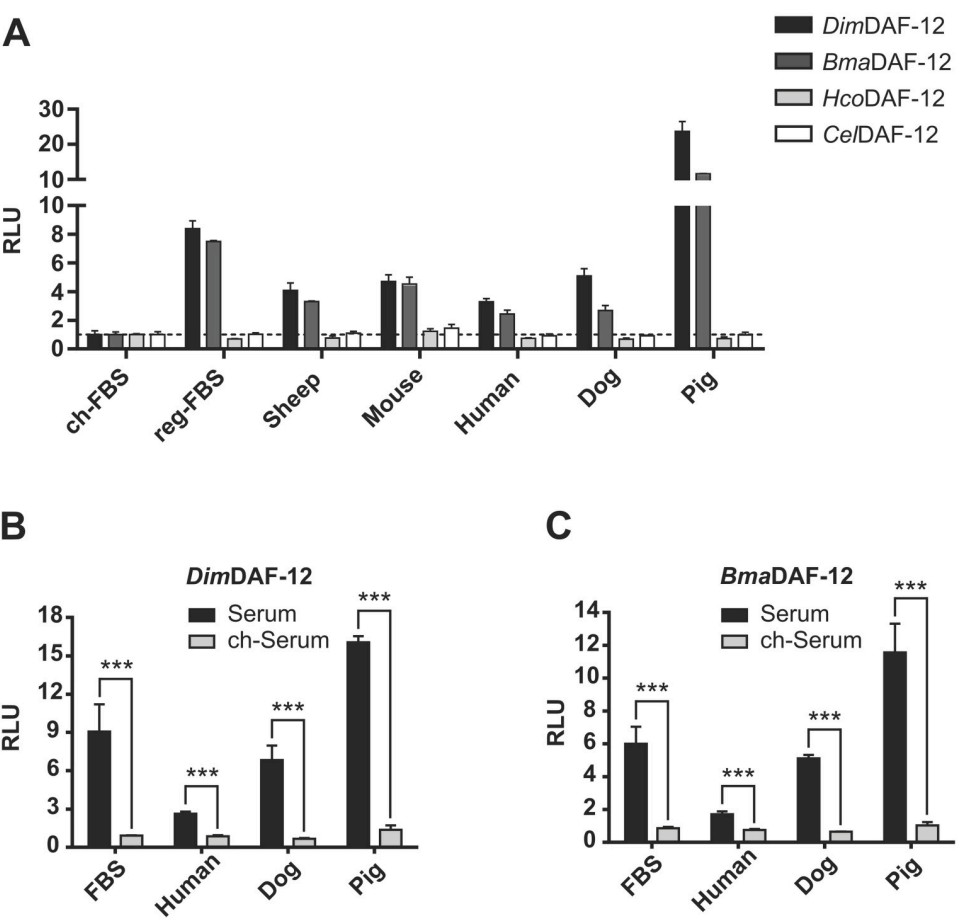

**Fig 4. Mammalian sera from different species activate both *Dim* and *Bma*DAF-12.** NIH3T3 cells were co-transfected with Gal4-DAF-12_LBD from *Dim*, *Bma*, *Hco* or *Cel* and luciferase gene reporter construct and then incubated for 24 hours with mammalian sera. (**A**) Activation of DAF-12 from *Dim*, *Bma*, *Hco* or *Cel* by regular sera from different species, compared with charcoal-stripped FBS, or (**B**) Activation of DAF-12 from *Dim* and (**C**) activation of DAF-12 from *Bma* by regular or charcoal-stripped sera from different mammalian species. Data represent the average of normalized luciferase activity and the error bars correspond to the standard deviations from three wells (A) or three independent experiments (B and C). The dash line in (A) shows an RLU of 1. The significance of the effects in (B) and (C) was analyzed by unpaired Student's t-tests using Prism 6.0 (Graph Pad Software, Inc.). ***p < 0.001.

*Bma*DAF-12 (Fig 4B and 4C), suggesting that the DAF-12 ligands are present in all these sera. Together, these results indicate that all mammalian sera may contain filarial DAF-12 ligands with similar steroid nature that can be removed by charcoal stripping, but their amount or their nature may differ between species.

## Δ4-dafachronic acid is a component of mammalian sera and participates in filarial DAF-12 activation

One of the best DAF-12 candidate ligands present in mammalian sera is cholestenoic acid (CA). CA has been quantified in human serum at approximately 190nM (Table 3) which is around its $EC_{50}$ for *Dim* and *Bma*DAF-12 (Table 2). However, in mouse serum CA is at insufficient concentration to activate DAF-12 (only 6nM to 8nM, Table 3). Thus, CA alone cannot explain the serum mediated DAF-12 activation. Therefore, there must be (an)other DAF-12 ligand(s), at least in mouse serum, which would account for its capacity to activate the filarial DAF-12. A careful reading of the literature to search for DAF-12 ligand candidates led to the unexpected discovery that mammalian sera contain Δ4-DA, also known as 3-Oxocholest-4-en-26-oic acid. Indeed, the recent development of an innovative approach for sterol analysis allowed the undeniable identification and quantification of Δ4-DA, in mouse and human sera (Table 3). We have listed a total of 6 publications that have quantified Δ4-DA with an average concentration of 4.1 nM and 3.8nM in human and mouse serum, respectively [18–23]. These concentrations are higher than its EC50 for *Bma* and *Dim*DF-12 (Table 2), which is more than sufficient to explain the serum-mediated activation of the filarial DAF-12. This supports the contention that Δ4-DA is the primary filarial DAF-12 ligands in mammalian sera. However, Δ4-DA has two stereoisomers at the C-25 position, (25S) and (25R), and the (25R)-isomer has been reported as a less potent activator of *Cel*DAF-12 than the (25S)-isomer. Therefore, we tested the potency of the (25R)-Δ4-DA isomer on *Bma* and *Dim*DAF-12 (Fig 5A). We observed EC50 around 5nM which is 5 times more than the commercial racemic formulation (50% R and 50% S) used in previous experiments (Fig 2A and 2B). This suggests that the filarial DAF-12 are less sensitive to the (25R)-Δ4-DA isomer than the (25S)-Δ4-DA isomer. Because (25S)-Δ4-DA is not commercially available we could not test its potency to activate individually filarial DAF-12 in our transactivation assay. These results imply that filarial DAF-12 activation by mammalian sera could be influenced by Δ4-DA stereo-chemistry. Moreover, which stereo-isomer is present in mammalian blood and in which proportion remain to be determined.

**Table 3. Review of the published research measuring the concentrations of Δ4-dafachronic acid and cholestenoic acid in human and mouse sera.**

| Concentrations in human serum (nM) | | | | |
|---|---|---|---|---|
| Δ4-dafachronic acid | | cholestenoic acid | | |
| **Mean** | Variability | **Mean** | Variability | **References** |
| **4.52** | 1.39 (SD) | **223.82** | 76.96 (SD) | Crick et al., *Mol Neurobiol*, 2017 [18] |
| **2.51** | 1.40–3.54 (range) | **132.01** | 105.92–154.14 (range) | Höflinger et al., *J Lipid Res*, 2021 [21] |
| **5.28** | 1.21 (SD) | **198.69** | 42.29 (SD) | Abdel-Khalik et al., *J Lipid Res*, 2017 [22] |
| **4.22** | 0.017 (technical SD) | **201.20** | 7.15 (technical SD) | Crick et al., *Clinical Chemistry*, 2015 [23] |
| Concentrations in mouse serum (nM) | | | | |
| Δ4-dafachronic acid | | cholestenoic acid | | |
| **Mean** | Variability | **Mean** | Variability | **References** |
| **3.43** | 1.14 (SD) | **6.42** | 3.00 (SD) | Crick et al., *JSBMB*, 2019 [19] |
| **4.21** | 0.18 (SD) | **8.55** | 2.93 (SD) | Griffiths et al., *BBA—MCB Lipids*, 2019 [20] |

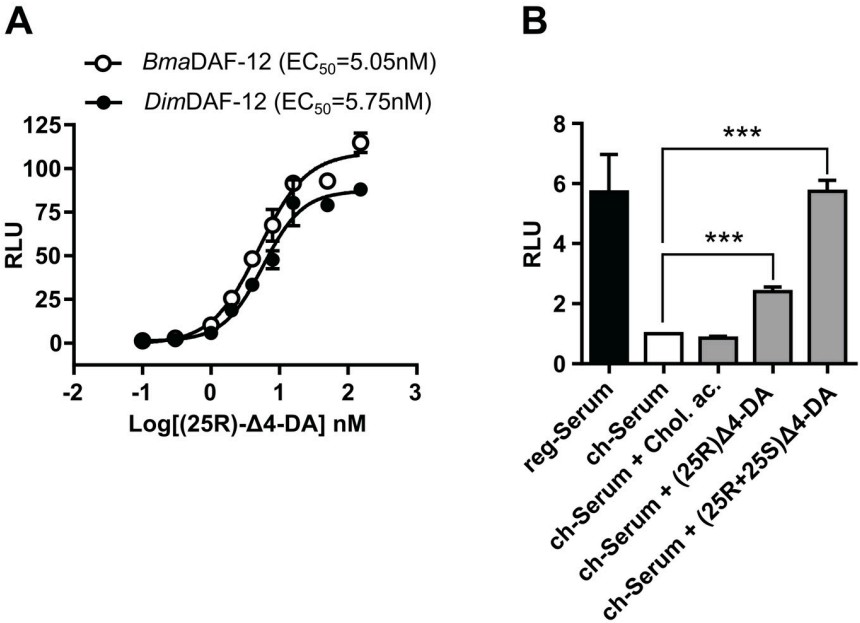

**Fig 5. Δ4-DA restores the activity of charcoal stripped mouse sera on *Dim*DAF-12.** **(A)** Dose-response curves of Δ4-DA (25R) on the activation of *Bma* and *Dim*DAF-12. NIH3T3 cells were co-transfected with Gal4-DAF-12_LBD from *Bma* or *Dim* and the luciferase gene reporter construct. Recombinant cells were incubated with increasing concentrations of Δ4-DA (25R). Data representing normalized luciferase activities are plotted with nonlinear regression fit using sigmoidal dose-response with variable slope (Prism 6.0, Graph Pad Software, Inc.). The figure shows one representative experiment out of two. Values are means of technical triplicates ± standard deviation. **(B)** NIH3T3 cells were co-transfected with Gal4-*Dim*DAF-12_LBD and luciferase gene reporter construct and then incubated with regular or charcoal-stripped mouse serum for 24 hours. Charcoal stripped mouse serum has been supplemented or not with the indicated DAF-12 ligands at 10% of the average concentration found in mouse serum and reported in Table 3 (0.75nM for cholestenoic acid (Chol. ac.) and 0.38nM for Δ4-DA). DAF-12 activity was normalized to the empty vector-transfected cells and to the charcoal stripped serum conditions and is expressed as relative light units (RLU). Data represent the average of normalized luciferase activity and the error bars correspond to the standard deviation of three independent experiments. The significance of the effects was analyzed by unpaired Student's t-tests using Prism 6.0 (Graph Pad Software, Inc.). ***p < 0.001.

To differentiate the relative importance of cholestenoic acid, (25R)-Δ4-DA and (25S)-Δ4-DA in the activation of filarial DAF-12 by mammalian sera, we spiked charcoal-stripped mouse serum with the different compounds at their average concentrations reported in the literature (Table 3 and Fig 5B). We calculated the concentrations in the assay, taking into account that only 10% serum was added in the cell medium: we used 10% of their average concentrations reported in the literature (Table 3 and Fig 5B). Charcoal stripping and addition of 0.75 nM of cholestenoic acid failed to activate *Dim*DAF-12. At the same time, (25R)-Δ4-DA at 0.38 nM partially restores the serum-dependent activation of *Dim*DAF-12, while supplying the charcoal-stripped mouse serum with the racemic Δ4-DA formulation led to activation of *Dim*DAF-12 comparable to the regular/unstripped mouse serum. These results provide evidence that Δ4-DA naturally present in the mammalian sera, participates significantly in filarial DAF-12 activation by the host serum.

## Dafachronic acids and mammalian serum accelerate the molting process of *D. immitis* infective third-stage larvae (iL3) into L4

We investigated whether DAF-12 ligands, naturally present in mammalian sera, could contribute to the development of *D. immitis* iL3. To this end, iL3 were extracted from mosquitoes and

after 1 day recovery in medium, they were incubated in medium supplemented with regular FBS, or charcoal-stripped FBS and exposed or not to Δ4-DA. Any larvae developmental changes relative to the molting process were recorded by measuring in parallel, (1) the shedding of the iL3 cuticle, which corresponded to L3 molting larvae (Fig 6A), and (2) the number of free cuticles in the medium, which was a sign that iL3 have completed the molting and have reached the L4 stage.

Addition of exogenous Δ4-DA in regular FBS containing medium clearly advanced the molting which started one day earlier than in absence of Δ4-DA (Fig 6B). This is in agreement with our previous observations showing that DAs and CA accelerate the molting process of *D. immitis* iL3 [14]. Interestingly, when iL3 were cultured with hormone-depleted serum (charcoal stripped), larval development slowed considerably while Δ4-DA restored the early

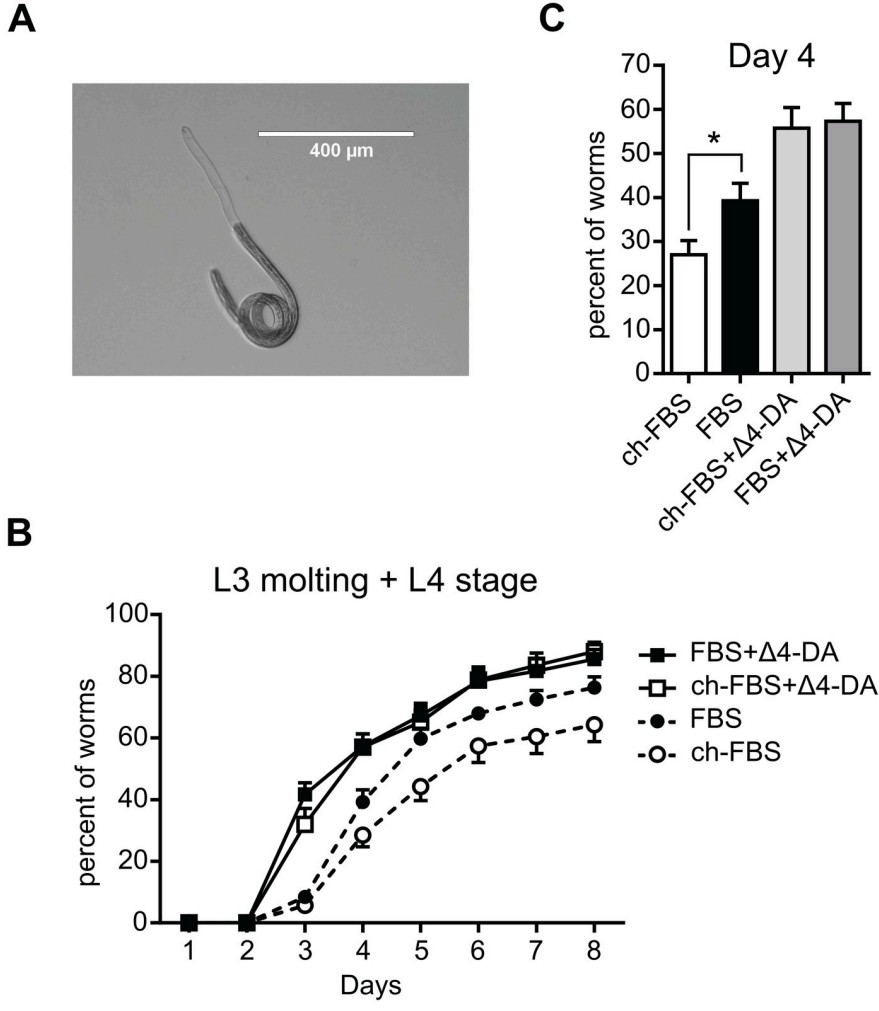

**Fig 6. Charcoal stripping of FBS affects the development of infective third stage D. immitis larvae.** (**A**) Image of a *D. immitis* L3 in process of completely shedding its cuticle (**B**) The iL3 stage larvae were collected from mosquitos and incubated in regular or charcoal-stipped FBS containing medium in the presence or absence of Δ4-DA. The iL3 undergoing molting or which had completed the molting from iL3 to L4 were scored daily under a microscope. The values plotted are the sum of the two effects: L3 molting + L4, and are the mean of 12 replicates with standard errors of the mean from 3 different pools of iL3s graphed using Prism 6.0 (Graph Pad Software, Inc.). (**C**) The percentage of developing L3 at day 4 extracted from the time course presented in (B). The significance of the effects was analyzed by unpaired Student's t-tests using Prism 6.0 (Graph Pad Software, Inc.). *p < 0.05.

development time as in regular FBS. Consistently, by comparing the random variables of a model of the percent of developing worms as a function of days (S4 Fig), we observed a significant effect of both exogenous Δ4-DA and serum charcoal stripping on T50 which is the time to reach half of maximum larval development (Emax), with respective p-values of 0.004 and 0.045. Interestingly, T50 increased from 4.0 days in regular FBS to 4.5 days in charcoal stripped FBS while it dropped to 3.4 days in the FBS + Δ4-DA conditions. Accordingly, at day 4 we observed 45% more development in FBS than in charcoal stripped serum (Fig 6C). Overall, these data suggest that exogenous Δ4-DA as well as ligands naturally present in mammalian serum sustain similarly the development of *D. immitis* iL3 through DAF-12 activation.

### RNAs encoding enzymes from the endogenous dafachronic acids biosynthesis pathways are down-regulated in the filarial nematode *Brugia malayi* at the time of infection

While we have observed that filarial DAF-12 are extremely sensitive to DA, we expected DA to be synthesized and present in filarial nematodes, since DAs have been quantified *in C. elegans* [24], *H. contortus* [15] and *S. stercoralis* [16], as well as in *Toxocara canis* and *Ascaris suum* [25]. DAs are produced in nematodes through specific biosynthesis pathways which encompass the enzymes DAF-36, DHS-16, HSD-1 and DAF-9 [26]. Indeed, DAF-9 have been clearly identified as the rate limiting enzyme of all branches of DA biosynthesis in *C. elegans* [11] and *S. stercoralis* [16]. However, DA synthesis pathways were unexplored in filarial nematodes. We identified the genes encoding potential homologs of DAF-36, DHS-16, HSD-1 and DAF-9 in the *B. malayi* genome (S1 Table), which share high homology with enzymes in *C. elegans*. Importantly, we found in the *D. immitis* genome a gene with 92% identity with the potential homolog of *daf-9* in *B. malayi*. This suggests that filarial nematode parasites could also produce their own DAs. However, the presence of a gene does not necessarily reflect its expression status, which is a prerequisite for any enzymatic activity.

In order to assess the DA synthesis pathways of filarial nematodes, we have exploited a publicly available transcriptomic dataset from *B. malayi*, across its entire *in vivo* life cycle [27]. We retrieved RNA sequencing read counts of DA synthesis enzymes and DAF-12 target genes (Fig 7A), from the iL3 to 8 days post-infection (dpi). Interestingly, in the iL3 stage at the time of infection, expression of *daf-36*, *dhs-16* and *hsd-1* were barely detectable (Fig 7B).

Strikingly, the expression of *daf-9* homologues is at almost undetectable levels at the time of infection, suggesting that production of DAs by the parasite, if at all, is low at the time of infection and during the early days of infection, relatively to other stages. This indicates that the filarial parasite *B. malayi* may not actively synthetize DAs at the initial infectious stages in the mammalian host. Importantly, the expression of *daf-12* is induced at the time of infection, as is *Lit-1*, a putative homologue of DAF-12 canonical target gene in *C. elegans* [28], suggesting that DAF-12 is ligand-activated, at a stage where the natural ligand produced by the parasite must be low or absent. Taken together these results suggest that while the ligand dependent activation of DAF-12 upon infection is an important step for filarial iL3 to L4 development, the parasite iL3 larvae may not produce significant amounts of endogenous DA, as the DA synthesis pathways at this stage may be switched-off. If this is the case, filarial nematode iL3 will rely essentially on the Δ4-DA present in the host blood to promote DAF-12 activation and provide a trigger for L3 development within the host.

## Discussion

The important role of DAF-12 in the development of the free-living nematode *C. elegans*, as well as of some parasitic nematodes such as *S. stercoralis* is well recognized, but there is still

**A**

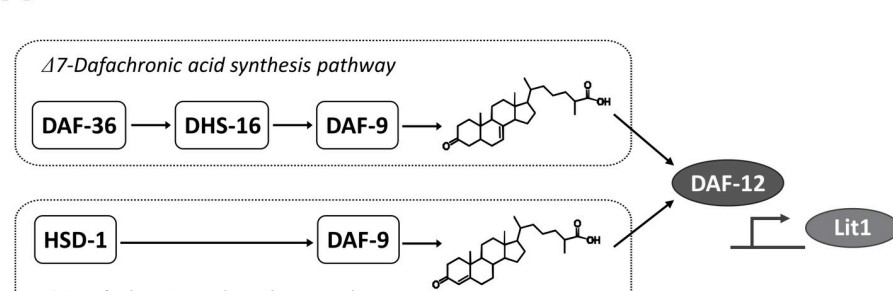

**B**

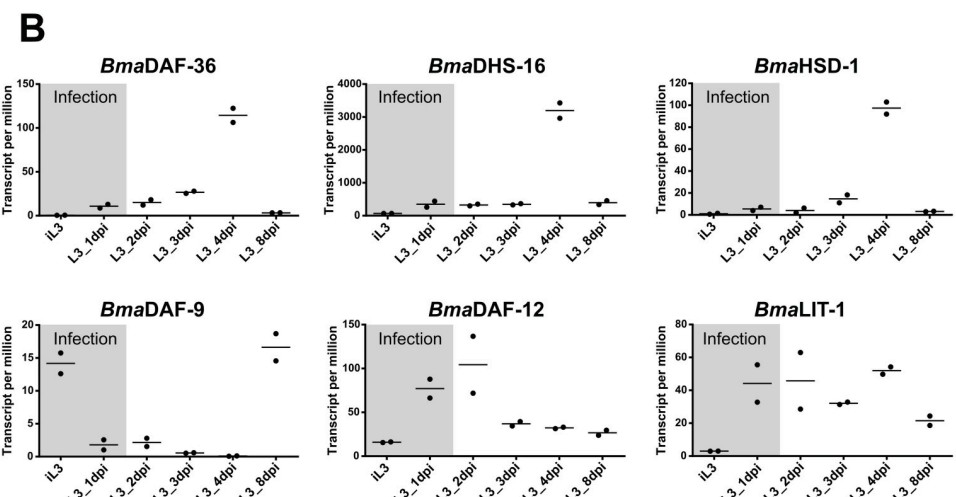

**Fig 7. Expression of homologue genes of the dafacronic acid synthesis pathways in *B. malayi*.** (**A**) Schematic representation of the Δ4-DA and Δ7-DA synthesis pathways with known enzymes. (**B**) Expression levels of the DA synthesis gene homologues *daf-36*, *dhs-16*, *hsd-1* and *daf-9* as well as *daf-12* and its target gene *lit-1* in *B. malayi* iL3 and L3 before and after infection of Mongolian gerbils. The Y-axis represents the relative gene expression levels given in transcript per million as analyzed by Chung, M et al. [27] (mSystems, 2019) from RNA-seq data. The variation in gene expression at time of infection is highlighted in grey. The data corresponding to two independent experiments are graphed.

much to be discovered, including in filarial nematodes. During larval development, DAF-12 acts as a molecular switch controlling quiescence entry and exit in nematodes [12]. Therefore, the binding of specific and natural ligands such as DAs to DAF-12 is the primary and most critical event of the dauer signaling pathway that decides between dauer entry or progression of the life cycle, and determines infection process in parasites. Recently, we have identified DAF-12 in *D. immitis* and showed that it is highly sensitive to DA activation, supporting that *D. immitis* possesses DAF-12-regulation pathway [14]. Intriguingly, the known DAF-12 ligands including DAs and CA, a steroid present in mammalian serum, exhibited much higher affinity for *Dim*DAF-12, than for orthologs from other nematode species. In order to determine if this is a specificity of *filarial* DAF-12, we first identified DAF-12 orthologues in *B. malayi*, a filarial nematode belonging to the clade III, and compared its activity with DAF-12 of *D. immitis*, and of *H. contortus* and *C. elegans*, from the clade V. Interestingly, *Bma*DAF-12 and *Dim*DAF-12 have both an exceptionnelly high sensitivity for DA, along with the capacity

to be specifically activated by mammalian sera. Interestingly, we have related such activation to the presence of Δ4-DA in mammalian sera, which also contributes to *Dim* iL3 development *in vitro*. Taken together, these findings reveal that filarial DAF-12 are able to sense Δ4-dafa-chronic acid in host serum to resume iL3 development upon infection of the mammalian host.

## First description of DAF-12 in *B. malayi*

We identified and cloned, for the first time, the orthologue of DAF-12 in *B. malayi*. By comparing sequences, activities and predicted structures of filarial DAF-12, with those of the nematodes *H. contortus* and *C. elegans*, we highlighted common and specific features of each species. They all displayed sequences of typical nuclear receptors, with highly conserved DBD, and more variable LBD. Based on LBD sequences, *Bma* and *Dim*DAF-12 were highly conserved, sharing 95% identity, while *Cel* and *Hco*DAF-12 exhibit only 58% identity. This reveals that both filariae receptors have similarly evolved, while some divergences occurred in the evolution of *Cel* and *Hco*. The specific selection pressures resulting from the life cycle of the filarial parasites such as invasive tissue stages of intermediate and definitive hosts may have shaped this evolution. The strong sequence identity of the filarial DAF-12 also predicts that they may share specific functional similarities. This aspect has been first explored *in vitro* using a cellular transactivation assay typically developed to assess the functioning of nuclear receptors [15]. Interestingly, we showed that *Bma*DAF-12 was highly sensitive to DAs and CA, which activated the receptor at very low concentrations, as previously demonstrated for *Dim*DAF-12, whereas much higher concentrations are required for activation of *Cel* and *Hco*DAF-12. It is noteworthy that Δ7-DA and Δ4-DA have similar high affinities for filarial DAF-12, whereas Δ4-DA is a much weaker ligand of *Cel* or *Hco*DAF-12 when compared to Δ7-DA. Interestingly, Δ7-DA have been detected in several nematode parasites including *H. contortus* [15] and *S. stercoralis* [16] while Δ4-DA has been, so far, quantified only in Clade III nematodes *T. canis* and *A. suum* [25]. These data reveal a clear functional species specificity of the filarial DAF-12 with a remarkable hypersensitivity for DAF-12 ligands, especially for Δ4-DA.

We previously pin pointed significant structural differences between *Dim*DAF-12 and receptors of Clade IV and Clade V in terms of residues implicated with ligand interactions [14]. For *Dim*DAF-12, the acid side of the ligands (e.g., DA) involves 2 salt bridges plus one H-bond, whereas they implicated 2 H-bonds and one salt bridge for *Ace* and *Sst*DAF-12. This implies a stronger affinity on this side since the bonding energy of a salt bridge is significantly higher than that of a hydrogen bond. Moreover, in contrast to *Ace* and *Sst*, *Dim*DAF-12 seems to have no H-bond on the 3-keto side which may render filarial DAF-12 ligand binding pockets more resilient for structural variations on this side of the ligands and may explain why CA, which exhibits a 3-hydroxy group, only activates filarial DAF-12. Such atomic details may account for the differential responses between receptors of filariae (Clade III) and Clade IV and V nematodes.

## Activation of filarial DAF-12 by mammalian sera

Another interesting finding was that *Dim* and *Bma*DAF-12 can be activated by hormonal compounds present in mammalian sera at low concentartions. Unexpectedly, *Dim* and *Bma*DAF12 but not *Cel* and *Hco*DAF-12, were partly activated in the cell-based transactivation assay by only 10% FBS, even though no ligand was added in the culture. This suggested that FBS contains some compounds which mediated the specific activation of filarial DAF-12. This was further supported by the dose dependent activation of *Dim* and *Bma*DAF-12 when the serum proportion was increased in the medium from 0% up to 20%. Not only FBS but also human, mouse, sheep, dog and pig sera were all able to activate *Dim* and *Bma* DAF-12. Interestingly,

charcoal-stripping completely abolished the ability of FBS, human, dog and pig sera to activate *Dim* and *Bma*DAF-12. Charcoal stripping is known to primarily remove steroid hormones from the serum [17] and is therefore commonly used to cultivate hormone sensitive cell lines. As mammalian sera contain CA, which is able to activate filarial DAF-12, we speculated that CA could be the primary filarial DAF-12 ligand of the serum. However, the relatively low potency of CA compared with DA, raises questions about its contribution to filarial DAF-12 activation by mammalian sera.

### Δ4-DA is present in mouse and human sera

Looking for mammalian compounds derived from cholesterol that could account for the activation of filarial DAF-12 by the mammalian sera, we were interested to find that Δ4-DA is present in human and mouse serum [18–23]. Indeed, when DA was characterized *in C. elegans* in 2006 [11] it was thought to be only produced by nematodes. In 2015 the development of quantitative charge-tags for the LC-MS analysis of oxysterols led to the discovery that Δ4-DA, known as 3-Oxocholest-4-en-26-oic acid, is produced by mammals [23]. This makes Δ4-DA the primary candidate for the activation of filarial DAF-12 by mammalian serum. Δ4-DA is one of the many mammalian oxysterols which are early intermediates in the metabolism of cholesterol to bile acids. Most of oxysterols possessing, as Δ4-DA, an acid tail are produced by the acidic bile acid synthesis pathway and are mostly of (25R) stereo-chemistry [29]. However, oxysterols (25S)-epimers have been identified in WT and CYP27A1-/- mice [20] and which one of the (25S) or the (25R) Δ4-DA is present in human and mouse serum is not known. To test which one of the DAF-12 ligands (cholestenoic acid, (25R)-Δ4-DA or (25S)-Δ4-DA) could contribute to activation of the filarial DAF-12 by the mammalian sera, we spiked charcoal stripped mouse serum with 10% of their average concentration reported in the litterature (as these cellular assays were performed in 10% serum). Addition of cholestenoic acid did not activate *Dim*DAF-12 at the concentration of 0.75 nM as expected from its high $EC_{50}$ for *Dim*DAF-12 (192nM). (25R)-Δ4-DA performed better with a partial restoration of mouse serum activity whereas the racemic formulation induced *Dim*DAF-12 activation to the same level as regular/ unstripped mouse serum. This is probably the consequence of the greater potency of the racemic formulation than the (25R)-Δ4-DA with $EC_{50}$ of 1.2nM and 5nM, repectively. This result provides evidence that regardless of its stereo-chemistry in the mammalian sera, Δ4-DA participates significantly in filarial DAF-12 activation by host serum. Since we have observed that the fiarial DAF-12 are specifically hyper-sensitive to Δ4-DA, this suggests that DAF-12 from filarial nematodes have evolved to sense the low concentration of Δ4-DA present in their mammalian host serum. Nevertheless, we cannot rule out that other steroids could also participate in filarial DAF-12 activation.

### Determinants of activation of *D. immitis* iL3 development

While Δ4-DA and mammalian sera activate filarial DAF-12, it was of interest that we showed that fetal bovine serum depleted of steroid hormones by charcoal stripping, negatively affected *D. immitis* iL3 development. We first confirmed the promoting effect of Δ4-DA on the development of iL3. Indeed, when Δ4-DA was added, iL3 development was accelerated and starting 1 day earlier than with serum alone. Moreover, addition of exogenous Δ4-DA accelerated iL3 development regardless of the serum used. By contrast, in the absence of exogenous Δ4-DA and with charcoal-stripped FBS, *D. immitis* iL3 development slowed, when compared to reference serum. The most significant change occurred at day 4, with 45% more *Dim* larvae undergoing development in the regular FBS than in charcoal stripped FBS (Fig 6C). This suggests that charcoal treatment removed a development stimulating factor. Knowing that DAF-12 is a

critical factor for quiescence exit in virtually all nematodes we propose that Δ4-DA naturally present in mammalian sera is directly sensed by filarial iL3 to stimulate their development. These findings suggest an adaptation of the dauer signaling pathway in filarial nematodes to rapidly exit quiescence and engage their development upon infection of the mammalian host. This could be particularly advantageous for filarial nematodes which enter their definitive host following the bite of an insect vector to rapidly adapt to a potentially hostile environment. Such rapid activation of the developing program may facilitate filarial nematode infection, in part by a prompt modulation of the immune system. Indeed, immune cells attracted to the wound may jeopardize nematode infection. Interestingly, it has been observed that the number of excreted/secreted proteins increases five times between L3 and molting-L3 in *B. malayi*, with several known immunomodulatory molecules of filarial parasites identified [30]. Importantly, thioredoxins which exhibit anti-inflammatory properties is solely secreted by the molting-L3 [30]. Moreover, relying on the mammalian host to provide the ligand hormone to initiate development of the iL3 larvae, rather than producing it endogenously in the filarial parasite would prevent precocious development while still in the mosquito intermediate host.

However, we cannot exclude that other mechanisms are also responsible for stimulating development. Indeed, the fact that charcoal stripping of the serum eliminates all DAF-12 activity in the transactivation assay, but did not completely abolish iL3 development, suggests the existence of other environmental factors which could promote larval development in vivo, possibly through stimulating endogenous filarial DA production. Consistently, elevated temperature mimicking the host body temperature stimulates endogenous DA synthesis and iL3 developmental resumption in *H. contortus* [15] and *S. stercoralis* [16]. While *Dim*iL3 only molt at 37˚C [31], this poses the question of whether filarial nematodes produce endogenous DAs during L3 development.

## Endogenous DA synthesis pathways supporting filarial specificity

The pattern of expression of the dauer signaling genes correlates with the capacity of *C. elegans* and *H. contortus* to produce endogenous DAs during quiescence exit and resumption of development [32,33]. Therefore, we sought to evaluate the expression profile of the enzymes involved in DA biosynthesis in filarial nematodes. We took advantage of an elegant transcriptomic study carried out *in vivo* for *B. malayi* at all stage of development [27]. The expression of the putative homologues of genes of the DA biosynthesis pathways was very low during the early days of infection in *B. malayi*, suggesting a low production of endogenous DA by the parasite. At the same time, the expression of *daf-12* as well as of the homologue of its putative target genes, *lit-1*, were high after infection, suggesting that the receptor is active. Based on RNA levels, it is hazardous to conclude on enzymatic activity of the product, since correlation between mRNA level and protein abundance is not mandatory. Nevertheless, the huge differences in RNA levels between the different stages and the coevolution of RNA encoding for strategic proteins belonging to DA synthesis/DAF-12 cascade in *B. malayi*, give original information on the global tendency of the gene expression program.

The low expression of the DA synthesis genes during the first days of the mammalian infection period raises the question of whether the filarial L3 larvae produce sufficient endogenous dafachronic acids to activate DAF-12. If the production of DA is low in L3 upon initial infection, DAF-12 activation would then depend mostly on the mammalian host Δ4-DA to sustain filarial parasites development. The expression of putative *B. malayi* homologues of *daf-36*, *hsd-1* and *dhs-16* peaked at 4 days post infection, and *daf-9* between 4 and 8 days, suggesting stimulation of endogenous production of DA a few days after infection. Accordingly, the first larvae molting occurred at day 4 in our *Dim* development assays. During dauer exit in *C. elegans*,

a DAF-12 dependent positive feedback loop facilitates development by ensuring that sufficient DA is dispersed throughout the body and serves as a robust fate-locking mechanism [34]. Similarly, filarial DAF-12 activation by the host Δ4-DA may feed such a feedback loop to stimulate endogenous DA production and ensure robust molting and development in the days following infection. This is consistent with observations made in *S. stercoralis* where DA levels become highest later in parasitic development within the host [16].

## Therapeutic strategies targeting DAF-12

Given the key role of DAF-12 activation in iL3 quiescence exit and development, it has been proposed as a promising drug target for treating parasitic diseases [9]. However, modulation of DAF-12 activity either for disease prevention or therapy will depend on the parasite life-cycle and specificities of the parasitic DAF-12. For instance, *S. stercoralis* is able to self-replicate within its host and possesses a unique autoinfection step in its life cycle which is initiated by auto-infective larvae. Remarkably, *Sst*DAF-12 functions as a molecular switch governing auto-infection and Δ7-DA administration suppresses autoinfection and markedly reduced lethality of the host in a gerbil model of strongyloidiasis [13]. While ivermectin is the treatment of choice for strongyloidiasis, it fails to eradicate the persistent auto-infective larvae. It is interesting that while Δ7-DA or ivermectin treatment causes a significant decrease of some development stages of *S. stercoralis* in gerbils, their combination results in the elimination of all stages [16]. Thus, ivermectin and Δ7-DA complement each other by targeting different stages of the lifecycle development. The situation must be different for filarial nematodes which do not exhibit an auto-infectious stage. Since we showed here that the direct activation of DAF-12 by the host Δ4-DA stimulates iL3 development, an appropriate strategy would be to inhibit DAF-12 to prevent this effect. Moreover, if some endogenous DA is produced by the larvae, a *bona fide* DAF-12 antagonist would inhibit both DAF-12 activation by exogenous and endogenous DA. Developing DAF-12 antagonists may be relevant to prevent filarial nematodes establishment as it would slow down the early larvae development. This should impair some crucial upstream steps of infection that are likely to alter the timely development of the parasite in the host. Importantly, chemoprophylaxis with MLs, including ivermectin, is the current standard preventive administrated to dogs and cats and this approach has proven its efficacy to prevent heartworm disease in endemic regions. However, resistances to MLs has been reported in the United States and its development is of concern in Europe [35]. Therefore, chemoprophylaxis with a DAF-12 inhibitor alone or combined with MLs could be a promising strategy to overcome the emergence of resistance to MLs because DAF-12 antagonists and MLs would have different mechanisms of action. Indeed, DAF-12 antagonists would alter the development of filarial larvae while MLs are thought to block the excretory-secretory apparatus of the parasites, thus preventing the secretion of immunomodulatory molecules [35,36]. Though caution should be taken in the development of DAF-12 antagonists as they may face cross reactivity with mammalian nuclear receptors such as FXR, the mammalian DAF-12 homolog which responds to bile acids or intermediates of the bile acids synthesis pathways. Fortunately, we have found that Δ4-DA does not activate FXR (S5 Fig) which suggests that the ligand binding pockets of DAF-12 and FXR are sufficiently divergent that it may be possible to find an antagonist that would target only DAF-12.

## Conclusion

Our data converge to show that filarial nematodes have evolved to sense very low concentrations of steroids present in mammalian serum. First, *daf-12* homologues are conserved in filarial nematodes, and DAs can activate *Dim* and *Bma*DAF-12 at very low concentration. Second,

even if filariae possess genes of the metabolic cascade required for DA synthesis, *B. malayi* larvae have very low expression of these genes at the stage of development corresponding to infection. We propose that upon infection of the mammalian host the Δ4-DA, contained in the serum, directly activates DAF-12 to readily trigger iL3 development, the starting point for adult parasite installation in the mammalian host. Given the importance of this specific step in the parasite's life cycle, targeting DAF-12 may represent an attractive opportunity to prevent filarial infections.

## Materials and methods

### Amplification and cloning

Total RNA from *B. malayi* L3 larvae was isolated using the Trizol method (Invitrogen). cDNA was obtained by reverse transcription using Superscript III first strand (Invitrogen). The full length *Bma*DAF-12 (Bm8452.1) was then amplified by High Fidelity Platinum Taq DNA polymerase (Invitrogen) using *Bma*DAF-12Fw/*Bma*DAF-12Rv primers (S2 Table) encompassing the predicted initiation and stop codon of the respective ORF based on the *B. malayi* genome reference Bmal-4.0. The product was subcloned into a pCR4 vector (Invitrogen) and the exact sequence was then confirmed by sequencing (McGill University/ Genome Quebec Innovation Centre). We found only one *Bma*DAF-12 cDNA sequence, which has been deposited in the GenBank database and is available under the accession number ON714136. We followed the same procedure to clone *Hco*DAF-12 cDNA using *Hco*DAF-12Fw/*Hco*DAF-12Rv primers (Supplementary file 3). The sequence of the cDNA was identical to the transcript ID HCON_00164850–00003.

### Plasmid constructs

To construct Gal4:DAF-12_LBDs, the LBDs of *Dim*, *Bma*, *Hco* and *Cel*DAF-12 were PCR amplified with the Phusion Hot Start II DNA polymerase High-Fidelity (Thermo Fischer) using the DAF-12-LBD-Fw and DAF-12-LBD-Rv primers listed in S2 Table. The PCR product were then digested with PvuI and EcoRI (New England BioLabs), for *Bma*, *Hco* and *Cel*DAF-12 LBDs or with PvuI and PmeI for *Dim*DAF-12. Following digestion, the PCR product were inserted into pFN26A vector (Promega), downstream of the DNA-binding domain of the Gal4 yeast transcription factor (Gal4-DBD). The sequences were then confirmed by sequencing (Eurofins Genomics).

### Cell-based transactivation assays

NIH3T3 cells (ATCC) were cultured in DMEM (Dulbecco's modified Eagle's medium) with L-glutamine supplemented with 10% (v/v) FBS containing 100 U/ml penicillin and 100 μg/ml streptomycin. To perform the cell-based transactivation assays, $12.5 \times 10^3$ NIH3T3 cells were seeded in white 96-well plates with transparent bottom. After 24 hours, the cells were transiently transfected in 125μl serum-free DMEM with 50 ng of pFN26A_DAF-12-LBD constructs bearing a Renilla luciferase gene used for normalization and 50 ng of pGL4.35 plasmid bearing the luciferase reporter gene under the control of the Gal4 response element (UAS, upstream activation sequence) (Promega) using 0.3 μl of TransIT-X2 Transfection Reagent (Mirus Bio) in 10μl of Opti-MEM. Serum-free medium was replaced 5 hours later by 200 μl of complete medium with ligands or vehicle control. The ligands used were: (25R+25S)-Δ4-dafachronic acid and (25S)-Δ7-dafachronic acid (Cayman Chemical) and 3β-hydroxy-5-cholestenoic acid (Santa Cruz Biotechnology) dissolved in DMSO. The final concentration of DMSO was maintained at 0.1% in each well. After 24 h incubation, the cells were lysed and luciferase

and renilla activities were successively measured using the Dual-Glo luciferase assay system (Promega) with a FLUOstar OMEGA microplate reader (BMG Labtech). In order to draw a dose response curve, we first calculated the ratio of luminescence from the luciferase reporter to luminescence from the renilla reporter for each condition and replicate. The ratio of DAF-12-transfected cells was normalized to the ratio of empty vector-transfected cells that were treated under the same conditions. The curve was fitted using GraphPad Prism 6 using a sigmoidal, 4PL, standard curve type (four-parameter logistic curve). Data presented for the cell-based transactivation assays are representative experiments (shown as mean data ± SD from triplicate).

## Animal sera and charcoal stripping

Human normal serum and regular or charcoal stripped fetal bovine serum (FBS), were obtained from Sigma-Aldrich. Sera from other animals were kindly provided by the engineers of the animal facilities at the Veterinary School of Toulouse for sheep and dogs, and at INRAE Toxalim research department for mice and pigs and were collected in polypropylene tubes during other experiments. All procedures to withdraw the blood were in accordance with European guidelines and regulations. Blood samples were incubated at room temperature to allow the blood to clot. The clots were removed by centrifuging at 2,500 x g for 5 minutes. The supernatants were transferred to new tubes and centrifuged again to remove any remaining blood cells. Sera were cleared using a 0.2μm filter and stored at -20˚C. For charcoal stripping, dextran-coated charcoal (Sigma-Aldrich) at 20g/L was rotated for 16hr at 4˚C in preparation buffer (0.25M sucrose, 1.5mM MgCl2, 10mM HEPES, pH 7.4) and centrifuged at 3,000 x g for 10 min. The supernatant was replaced with the same volume of serum and incubated for 16hr at 4˚C. After centrifugation to remove the dextran-coated charcoal, the serum was cleared using a 0.2μm filter and stored at -20˚C.

## Development of *D. immitis*

*D. immitis* development assay has been performed essentially as described previously [14]. Briefly, the laboratory-maintained *D. immitis* (2005 Missouri strain) was obtained from the Filariasis Research Reagent Resource Center (FR3) [37]. The iL3 were collected from infected mosquitoes, washed in PBS and incubated in RPMI-1640 with L-glutamine (Gibco) containing 10% FBS and antibiotics. The next day, the iL3 were transferred to a 24-well plate (15–18 per well) and maintained at 37˚C under 5% CO2 for 9 days, with fresh complete RPMI-1640 containing 10% regular FBS or charcoal FBS, with or without 10 μM of Δ4-DA. The medium was renewed every two days and the worms were observed every 24 h under a stereomicroscope. The molting from the iL3 to the L4 larvae was recorded by scoring the number of larvae which have initiated but not completed the molt, and the number of free cuticles into the medium, as a reflect of the number of L4 that have completed the molting. The percentage of iL3 engaged in the molting process and of L4 were calculated every day as follows: number of molting iL3 or L4 in the well /initial number of larvae per well× 100.

## Identification of *C. elegans* gene homologues in *B. malayi*

We identified homologs of the DA biosynthetic pathways by searching (tblastn; e-value: $\leq 10^{-20}$) the *C. elegans* protein sequences against gene predictions from the latest, published genomes of *B. malayi* (Bmal-4.0) on wombaseparasite (parasite.wormbase.org). To confirm that these genes could be potential homologs we perfomed a reverse blast by using *B. malayi* protein sequence against genes from the genome of *C. elegans* (tblastn; e-value: $\leq 10^{-20}$).

## Data analysis

The dynamic of the percentage of developing worms with time was analyzed with the following nonlinear mixed effects model:

$$Y_{ijkl} = \frac{Emax_{ij}t_k^{\alpha_{ij}}}{T50_{ij} + t_k^{\alpha_{ij}}} + \varepsilon_{ijkl},$$

where $Y_{ijkl}$ is the percentage of worms observed in the $l$th well at time $t_k$, with (and without) serum charcoal stripping corresponding respectively to i = 1 and 2 and, with (and without) Δ4-DA corresponding respectively to j = 1 and 2. The effect of presence/absence of serum and DA as well as their interactions were respectively tested for the random variables $Emax_{ij}$, $T50_{ij}$, $\alpha_{ij}$ with a likelihood ratio test (LRT). The data were analyzed using the library *nlme* of the R software.

## Supporting information

**S1 Fig. Sequence alignment of the LBD from *Cel*, *Hco*, *Bma* and *Dim*DAF-12.** Multiple sequence alignment was performed with the MAFFT program from MyHits. Identical (black) and similar (grey) residues have been colored coded with the Color Align Conservation program.
(TIFF)

**S2 Fig. Expression of DAF-12_LBD from different nematode species fused to the DBD of GAL4 in NIH3T3 cells.** Western-blot analysis of whole cell lysate of NIH3T3 transfected with the pFN26A plasmid carrying either GAL4, GAL4-*Cel*DAF-12, GAL4-*Hco*DAF-12, GAL-4-*Dim*DAF-12 or GAL4-*Bma*DAF-12 using GAL4 antibody and Lamin A antibody for loading control.
(TIF)

**S3 Fig. Mammalian sera from different species and multiple donors activate *Dim*DAF-12.** NIH3T3 cells were co-transfected with Gal4-*Dim*DAF-12_LBD and luciferase gene reporter construct and then incubated for 24 hours with sera from different mammalian species from different individuals. Data represent the average of normalized luciferase activity and the error bars correspond to the standard deviations from three wells.
(TIF)

**S4 Fig. Fitting curves from a nonlinear mixed effects model of the percent of developing worms as a function of days corresponding to the data presented in Fig 6B.**
(TIF)

**S5 Fig. Δ4-DA and CA do not activate FXR.** NIH3T3 cells were co-transfected with Gal4-FXR_LBD and the luciferase gene reporter construct before incubation with DMSO (0.1%) or with 10μM of Δ4-DA, CA or of the synthetic FXR agonist GW4064 for 24 hours. FXR activity was normalized to the empty vector-transfected cells and DMSO and expressed as relative light units (RLU). Data represent the average of normalized luciferase activity and the error bars correspond to the standard deviation of triplicates.
(TIF)

**S1 Table. DA synthesis enzyme genes in B. malayi genome.**
(DOCX)

**S2 Table. DNA Primers used in this study.** * Tm have been calculated with the NEB Tm calculator for the Phusion Hot Start Flex DNA polymerase Buffer. For oligos containing

restriction sites for cloning, Tm values for full length as well as for the sequence identical to the target are given.
(DOCX)

## Acknowledgments

We thank the NIH/NIAID Filariasis Research Reagent Resource Center (FR3) (www.filariasiscenter.org) for distribution by BEI Resources, for providing the following reagent: Stage iL3 *Dirofilaria immitis*, Strain Missouri, Infective Larvae (Live), NR-48909. We thank Marie Garcia for her technical supports and Mélanie Alberich and François André for their fruitful discussions. We thank Beatrice Roques, Philippe Pinton, Joelle Laffitte and Alix Pier-ron for providing animal sera.

## Author Contributions

**Conceptualization:** Rémy Bétous, Roger Prichard, Anne Lespine.

**Data curation:** Rémy Bétous, Anthony Emile, Roger Prichard, Anne Lespine.

**Formal analysis:** Rémy Bétous, Anthony Emile, Eva Guchen, Didier Concordet.

**Funding acquisition:** Roger Prichard, Anne Lespine.

**Investigation:** Rémy Bétous, Anthony Emile, Hua Che, Eva Guchen, Didier Concordet, Thavy Long.

**Supervision:** Rémy Bétous, Roger Prichard, Anne Lespine.

**Writing – original draft:** Rémy Bétous.

**Writing – review & editing:** Rémy Bétous, Sandra Noack, Paul M. Selzer, Roger Prichard, Anne Lespine.

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
