## [Decision Letter · Decision Letter 0]

25 Oct 2022

Dear Dr Betous

Thank you very much for submitting your manuscript "Filarial DAF-12 sense Δ4-dafachronic acid in host serum to resume iL3 development during infection" for consideration at PLOS Pathogens. As with all papers reviewed by the journal, your manuscript was reviewed by members of the editorial board and by several independent reviewers. In light of the reviews (below this email), we would like to invite the resubmission of a significantly-revised version that takes into account the reviewers' comments.

Thank you for submitting your manuscript to Plos Pathogens. Your manuscript has been reviewed by 3 experts in the field. There is interest in your manuscript and the results presented, but there is a need for additional experiments to determine more clearly if the activating molecule is delta4-dafachronic acid; there is concern that sera from multiple non-permissive hosts (including FBS) stimulate DAF-12, possibly suggesting that the activating effect of the serum is not a host determining factor. Evidence to support the conclusion that the host serum factor that activates filarial DAF-12 species is Δ4-DA is not directly supported by the data presented in the paper. It is recommended that you review carefully the critiques of the referees, suggestions for additional experiments and address each of the comments raised by the referees before the manuscript can be considered further.

We cannot make any decision about publication until we have seen the revised manuscript and your response to the reviewers' comments. Your revised manuscript is also likely to be sent to reviewers for further evaluation.

Sincerely,

Richard J. Martin, BVSc, PhD, DSc, DipECVPT, FRCVS

Guest Editor

PLOS Pathogens

P'ng Loke

Section Editor

PLOS Pathogens

Kasturi Haldar

Editor-in-Chief

PLOS Pathogens

orcid.org/0000-0001-5065-158X

Michael Malim

Editor-in-Chief

PLOS Pathogens

orcid.org/0000-0002-7699-2064

Thank you for submitting your manuscript to Plos Pathogens. Your manuscript has been reviewed by 3 experts in the field. There is interest in your manuscript and the results presented, but there is a need for additional experiments to determine more clearly if the activating molecule is delta4-dafachronic acid; there is concern that sera from multiple non-permissive hosts (including FBS) stimulate DAF-12, possibly suggesting that the activating effect of the serum is not a host determining factor. Evidence to support the conclusion that the host serum factor that activates filarial DAF-12 species is Δ4-DA is not directly supported by the data presented in the paper. It is recommended that you review carefully the critiques of the referees, suggestions for additional experiments and address each of the comments raised by the referees before the manuscript can be considered further.

Reviewer's Responses to Questions

**Part I - Summary**

Reviewer #1: This manuscript, entitled “Filarial DAF-12 sense Δ4-dafachronic acid in host serum to resume iL3 development during infection”, describes the characterization of DAF-12 orthologues isolated from B. malayi and D. immintis in vitro. The authors made several interesting observations, including that the filarid DAF-12 ligand binding domains (LBDs) are more sensitive to Δ4-DA than those from C. elegans and H. contortus, and present evidence that iL3 detect and are stimulated by host DAs during infection. The manuscript is interesting and makes some provocative conclusions that are not completely supported by the data. There are several minor and major issues with the manuscript that preclude a recommendation for publication. Specifically:

1. A major conclusion is that DAs are directly sensed by D. immitis L3 and are determinants of activation of the larvae. However, the data clearly show that DA is a relatively minor contributor to activation and are unlikely to be the major signal that initiates development, as significant molting and development occurs in charcoal stripped serum. This is also supported by the non-specificity of serum to activate DAF12 and to promote development. Sera from multiple non-permissive hosts (including FBS) stimulate DAF-12, suggesting that the activating effect of the serum is not a host determining factor. The authors do not show the effects of sera from other species that bind and activate DAF-12. For example, pig serum activates DAF-12 in cell culture, but were not tested for its effect on molting and development. This is an important missing experiment.

2. Perhaps equally interesting is what other stimuli are present in stripped sera. Stripping serum is likely removes more than just sterols and DA. The potential roles of the remaining serum components are unknown, and not addressed by the authors.

3. What is the effect of Δ4-DA alone on molting? The authors referenced a previous study, but this control should be included in the current experiments.

4. The experiments reported in figure 6 are intriguing, and the interpretation may be correct, but transcript levels do not always reflect protein levels. Does inhibition of DA synthesis prevent molting and/or development? This would confirm whether DA synthesis was required during infection or if host sterols are the initiating factor.

5. Similarly, on Line 411 the authors pose the question of whether filarial nematodes produce DA during L3 development. Also, on line 426 they suggest that the parasite does not actively produce endogenous DA. This could be tested with DA synthesis inhibitors, i.e. do inhibitors prevent the development stimulated by serum or stripped serum?

6. Line 439: One might actually conclude that other signaling pathways are involved given the relatively minor effect of DA on serum stimulated development.

7. Line 467: The authors need to justify how DAF-12 antagonists would prevent establishment given that worms develop without its ligand.

8. Table 2. There are insufficient details regarding how the EC50s are calculated. The authors should include the SE of the logEC50 and the number of biological and technical replicates in the table.

Minor:

1

Reviewer #2: This manuscript posits some intriguing hypotheses on the mechanism(s) by which filarial iL3 activate inside the mammalian host; however, it seems to suffer from over-interpretation of the data presented. I would strongly recommend additional experiments (outlined below) in addition to revisions of the interpretations.

Reviewer #3: This manuscript characterizes the nuclear receptor DAF-12 in filarial nematodes and claims that host serum contains a dafachronic acid (Δ4-DA) that directly activates the parasite DAF-12, which in turn promotes the parasite’s L3 quiescence exit. The authors show that DAF-12s isolated from B. malayi and D. immitis, two prevalent filarial nematodes, are activated by dafachronic acids, which is not a particularly novel or surprising discovery since these compounds have been previously shown to bind and activate DAF-12 from several other free-living and parasitic nematodes. The apparent novelty of the paper is demonstrating that sera from multiple mammals can activate the bm- and di-DAF-12s and promote filarial nematode development. Since an isomer of Δ4-DA has previously been reported to be present in human serum, the authors concluded that Δ4-DA is the physiological signal in serum that promotes filarial nematodes to exit L3 quiescence for reproductive development. The authors also profiled the mRNA expression of the homologs of known DA biosynthetic enyzmes to conclude that worms do not make DA. Unfortunately, neither of these conclusions are well supported by the data in its present form.

**Part II – Major Issues: Key Experiments Required for Acceptance**

Reviewer #1: 1. Does serum from other species (e.g. pig) induce molting and development?

2. Effect of delta 4 alone on molting.

3. Effect of DA synthesis pathway inhibitors on development/

Reviewer #2: 1) C. elegans has seven daf-12 isoforms, S. stercoralis has three daf-12 isoforms, and D. immitis has two daf-12 isoforms. Is there more than one daf-12 isoform in either B. malayi or H. contortus? Is there any conservation of daf-12 isoforms between clade III, IV, and V nematodes? Which isoform was cloned and used in this study? These are pertinent questions that should be answered.

2) In Figure 1, why are the ligand binding domains aligned separately for B. malayi & D. immitis (panel B) and then for C. elegans & H. contortus (panel C)? Why not align all of the LBDs together in a single panel, similar to A? Also, the white amino acid letters are difficult to read on the black background.

3) Lines 128-129: Without an experiment to test this, I do not think the authors can claim that the DAF-12 DNA-binding domain has a “similar binding mode of the receptors on promoter binding sequences.” There are three amino acid differences in the DNA-binding domain and the impact these have on recognizing target sequences is unknown.

4) Figure 2. The absolute RLU for Hco and Cel constructs is nearly twice as high as the RLU for Dim and Bma constructs. It is be important to show that similar levels of fusion protein are being made for these four constructs; presumably they were not codon optimized for expression in NIH/3T3 mammalian cells and thus some of the apparent differences in activity could actually be a result of different levels of protein expression. An immunoblot would be an important confirmation step. Additionally, please make it clear in the figure that these are ligand binding domains (e.g., DimDAF-12-LBD), and in the figure legend that these are a fusion proteins of the GAL4-DBD with the nematode LBD.

5) Figure 3. It is really unclear how physiologically relevant the activation of DimDAF-12 LBD and BmaDAF-12 LBD by FBS is. The assay is the same as in Figure 2, yet the RLU (which maxes out at 8) are approximately at the level of detection for the presumed endogenous dafachronic acid ligands (which max out at 80-100 RLU). These data indicate that the steroid hormones in non-stripped FBS are not native ligands for DimDAF-12 or BmaDAF-12. These experiments need to be repeated with the additional control: charcoal-stripped FBS + cholesterol (Hoflinger et al. 2021 reports ~2 mg/ml total cholesterol in human serum). If the marginal activation of the Dim and Bma DAF-12 LBDs is due to non-specific binding of sterols, cholesterol would presumably active them similar to FBS; contrarily, if there is a sterol that is a relevant ligand for Dim and Bma DAF-12 LBDs, which is just at a very low concentration in FBS, then cholesterol should be similar to charcoal-stripped FBS.

6) Figure 4. Was the serum from various mammalian species from a single individual? A pool of individuals? Or each replicate contained serum from a different mammalian donor? I am not certain that this experiment takes into account relevant biological variation between individual mammalian organisms.

7) Lines 245-248. I am uncertain how relevant Emax is for this experiment, since the experiment was stopped after 8 days, and the percentage of larvae molting was continuing to increase between days 7 and 8 for all conditions. I recommend removing the references to Emax, unless this experiment is repeated and the number of days extended for several additional days until no further increases in activation are observed.

8) Line 260 and Lines 569-572: The homologs of daf-36, dhs-16, hsd-1, and daf-9 should be identified using both forward and reverse BLAST searches. I.e., the top hits of C. elegans queries should be used to search the B. malayi genome, and the predicted B. malayi proteins should be used as queries to search the C. elegans genome. The authors need to determine whether these are one-to-one homologs or merely genes encoding that particular family of proteins. Any discrepancies should be resolved with phylogenetic analyses. These experiments need to be performed and described.

9) Figure 6 and Line 277: Standard deviation cannot be calculated from only two replicates (minimum of three biological replicates required). Please correct this in the text and the figure; I would recommend simply plotting both data points with no error bars. Since biological variance cannot be calculated using two replicates, it is not possible to determine any statistically significant differences. Also, please provide either a reference or evidence for B. malayi lit-1 as a direct target of B. malayi DAF-12.

10) Lines 279-288. The authors appear to assume that the presence or absence of a particular transcript directly correlates with the presence or absence of the protein. However, in studies of other parasitic nematodes with paired RNA-Seq and mass-spec data for the same developmental stage, there is little correlation between mRNA abundance and protein abundance. This is particularly anticorrelated for iL3, which store both proteins and mRNAs. Without any data to support or contradict DAF-9 activity in iL3, I would recommend against making any statements about whether or not iL3 produce dafachronic acids upon encountering a host.

11) Lines 288-290 and 365-375. If filarial iL3 rely on delta4-dafachronic acid (formal name: 3-oxo-cholest-4-en-26-oic acid) for their activation, and Hoflinger et al. 2021 (ref. 17) identified 3-Oxocholest-4-en-(25R)26-oic acid at a concentration of ~3 nM in human serum, why doesn’t human serum activate either DimDAF-12 LBD or BmaDAF-12 LBD (~3 RLU in Fig. 4) even close to the extent that delta4-dafachronic acid does (at EC50 1.2 – 1.7 nM; Table 2; ~50 RLU in Fig 2)?

12) Overall, the authors do not provide any data supporting the assertion that the activating molecule in FBS is indeed delta4-dafachronic acid. The data presented are only correlative and the conclusions speculative. Much more convincing would be: HPLC/MS data showing that delta4-dafachronic acid is indeed in FBS (and at what concentration), HPLC/MS data showing that charcoal stripping removes delta4-dafachronic acid from FBS, and that charcoal-stripped FBS + synthetic delta4-dafachronic acid (at the concentration found by HPLC/MS) activates DAF-12 with similar efficacy as FBS. This set of experiments would be much more convincing!

Reviewer #3: 1. Evidence to support the conclusion that the host serum factor that activates filarial DAF-12 species is Δ4-DA is not directly supported by the data in this paper. Indeed, the authors present no evidence that serum contains Δ4-DA at concentrations that would activate the receptor and affect worm development. Instead, the authors cite the key previous paper (Höflinger, J Lipid Research, 2021) as evidence that serum contains Δ4-DA. However, a careful reading of that paper reveals that this Δ4-DA compound, also known as 3-oxocholest-4-en-(25R)26-oic acid, is actually the 25(R) enantiomer of Δ4-DA. In that paper, the reported concentration was 1nM. A key missing experiment is for the authors to show in their samples that the active form of Δ4-DA is present, which they did not do. Even if this compound is in the serum sample that the authors used, there are two things that still do not support their conclusion. First, the pharmacological properties of dafachronic acid ligands for DAF-12 receptors depend on the stereochemistry at the C-25 position, i.e., the (S) versus (R) form. The 25(S)-DA is at least an order of magnitude more potent than the 25(R)-DA in activating all other tested DAF-12 receptors. As noted above in the cited reference, only the 25(R) enantiomer was found, which makes sense because unlike the DAF-9 enzyme in nematodes, the 26-hydroxylase in mammals that catalyzes the 26-oxdidation favors 25(R) products (Javitt, J. Lipid Res. 1990). A direct comparison of the activities of the two Δ4-DA enantiomers is needed to make any conclusions on that point. More importantly, to demonstrate what the activity is in the serum used in these studies, the authors should chemically isolate and identify the compound in their own serum samples, which might be quite different than those reported in the cited study. The results of that experiment would be of great interest and directly address the main conclusion of the paper.

2. The authors show that mammalian sera can activate the D. immitis DAF-12 (Fig 4) and promote L3 quiescence exit (Fig 5). They use these experiments as further evidence to support the notion that the serum activity is Δ4-DA. However, again there is no direct evidence to support this. Importantly, while charcoal stripping essentially eliminated all DAF-12 activation, it only marginally impaired L3 quiescence exit (Fig. 5), which should not have been the case if this was due to the lack of Δ4-DA. It also is notable that whatever this activity is in the transfection assay, it is much weaker than Δ4-DA. In thinking about this more, a good candidate for this activity might be cholestenoic acid or one of the other congeners that are found at much higher concentrations in serum as reported in the Höflinger, J Lipid Research paper. Cholestenoic acid is at a concentration in serum that is in the range of activity that the authors found in Fig. 2. Given that cholestenoic acid is also a poor activator of C. elegans DAF-12, this would also explain why serum did not activate C. elegans DAF-12, while it showed some activity on the filarial DAF-12s. Furthermore, since adding exogenous Δ4-DA to FBS can accelerate the L3 quiescence exit (1 day earlier, Fig 5B), it is equally if not more possible that the serum stimulates DA synthesis in the parasite, which takes some time. Taken together, all these findings further support the alternative conclusion that something in serum other than Δ4-DA may be directly, but weakly activating DAF-12, and/or that serum is also stimulating the true endogenous ligand’s synthesis. Again, without isolating the compound and showing its identity, it is not possible to draw any conclusions.

3. A third piece of evidence presented to support the authors’ conclusion that the parasites do not synthesize DA is that the mRNA expression of a protein the authors annotate as “BmaDAF-9” is very low during the L3 quiescence exit. This however does not prove DA synthesis in the parasites does not exist. There are a couple of problems with this interpretation. First, the authors provided no conclusive evidence to show whether the “BmaDAF-9” claimed in this study is the DAF-9 ortholog. The authors make this claim based solely on bioinformatic analysis without any experimental evidence. Indeed, the protein sequence identity compared to C. elegans DAF-9 is only ~35% (by both Blastp of NCBI and Clustwal Omega of EMBL-EBI), which is too low to predict what it might do. In fact, the very logic of this strategy seems contradictory: if the authors are claiming “BmaDAF-9” is the daf-9 gene (as identified by homology) then it should function as DAF-9. As recently demonstrated in Strongyloides stercoralis, one cannot conclude the identity of the DAF-9 ortholog without testing its activity in a biochemical assay (Wang, eLife 2021). The real experiment that needs to be done is to test all of the potential cytochrome P450s in the parasite for their ability to make DA (using as substrates either 4-cholestene-3-one or lathosterone). Lastly, mRNA levels do not necessarily correlate with protein levels. This is especially true for DAF-9 where a feedback regulation of its mRNA levels by DAF-12 activation has been reported in C. elegans (Wang, Plos Genetics, 2015). A simple way to test whether these parasites synthesize DA might be to feed the parasites isotope-labeled cholesterol and then assay the incorporation of the isotope into DA has been done recently in Strongyloides stercoralis (Wang, eLife, 2021).

**Part III – Minor Issues: Editorial and Data Presentation Modifications**

Reviewer #1: 1. Fig. 1. Why are all the LBDs not included in the same figure?

2. Figure 5 legend, lines 232-234: Are the pools individual biological replicates? Please clarify.

3. Numerous typos. Some of them are listed here on lines:

• 249: similarly

• 318: receptor

• 347: concentration

• 394: activate

• 396: rapidly

• 416: constantly

• 418: Cel-daf-9

• 491: mosquito

Reviewer #2: Lines 43: It is probably best not to anthropomorphize the parasites (“do not seek to produce”).

Line 60: What would be the difference between a “pre-L1” and a presumed “L1”?

Line 64-65: What about global prevalence?

Line 73: Clarify that DAF-12 is only relevant to nematode pathogens, since daf-12 has only been found in nematodes.

Lines 79-81: This sentence needs a reference.

Line 96: beta symbol is missing.

Line 167: Without mass-spec data from ground-up worms to really show that delta4-DA and delta7-DA are truly the endogenous ligands for DimDAF-12 and BmaDAF-12, this sentence should be qualified.

Line 178: DAF-12 ligand binding domains.

Figure 5D: Please put error bars both above and below each of the symbols. Also, what temperature were larvae incubated at?

Lines 235-236: Is this difference statistically significant? If so, please state this, the p-value, and test used.

Line 257: The role of HSD-1 in the synthesis of DA in parasitic nematodes is unclear, but does seem to play a role in P. pacificus (doi: 10.1093/genetics/iyab071).

Line 259-260: genes encoding potential homologs

Figure 6: The words in panel B are unreadable. Please use a higher quality image.

Lines 333-335: These data are presented for the first time in the Discussion. Please move to the Results.

Methods: please include CAS numbers for compounds

There are several spelling and grammatical errors in this document, which would benefit from copy editing prior to publication.

Reviewer #3: N/A

PLOS authors have the option to publish the peer review history of their article (what does this mean?). If published, this will include your full peer review and any attached files.

Reviewer #1: No

Reviewer #2: No

Reviewer #3: No
---

## [Decision Letter · Decision Letter 1]

25 May 2023

Dear Dr Betous,

Thank you very much for submitting your manuscript "Filarial DAF-12 sense Δ4-dafachronic acid in host serum to resume iL3 development during infection" for consideration at PLOS Pathogens. As with all papers reviewed by the journal, your manuscript was reviewed by members of the editorial board and by several independent reviewers. The reviewers appreciated the attention to an important topic. Based on the reviews, we are likely to accept this manuscript for publication, providing that you modify the manuscript according to the review recommendations.

The revised manuscript and response to the referees has been carefully reviewed by the editors. The manuscript is significantly improved but there remains concerns (described fully by referee 3) about the justification of key conclusions. These concerns may be best addressed in a revised discussion.

Sincerely,

Richard J. Martin, BVSc, PhD, DSc, DipECVPT, FRCVS

Guest Editor

PLOS Pathogens

P'ng Loke

Section Editor

PLOS Pathogens

Kasturi Haldar

Editor-in-Chief

PLOS Pathogens

orcid.org/0000-0001-5065-158X

Michael Malim

Editor-in-Chief

PLOS Pathogens

orcid.org/0000-0002-7699-2064

The revised manuscript and response to the referees has been carefully reviewed by the editors. The manuscript is significantly improved but there remains concerns (described fully by referee 3) about the justification of key conclusions. These concerns may be best addressed in a revised discussion.

Reviewer Comments (if any, and for reference):

Reviewer's Responses to Questions

**Part I - Summary**

Reviewer #3: This revised manuscript is improved, but still does not satisfactorily justify a couple of the key conclusions of the paper. The claim in the title that “filarial DAF-12 sense Δ4-dafachronic acid in host serum to resume iL3 development” is overstated and based on circumstantial, and in some cases, contradictory evidence. While it is possible that this statement is true or partially true, this definitive statement was not proven and is based on circumstantial evidence. In addition, the authors ruling out endogenous DA synthesis as not being required is also not justified. The current data and conclusions fail to take into account other possibilities that were not ruled out as explained below:

1) A compound other than D4-DA may be present that activates development. Although the authors provide published data from others that some form of DA may be present, the fact that serum stripping eliminates all DAF-12 activity but still induces development continues to be convincing evidence for the existence of another mechanism. The assay the authors used to monitor DAF-12 activity is robust and highly sensitive for detecting even low levels of DA, so the fact that charcoal stripping removes all activity strongly suggests there is no significant ligand left in the serum. Nevertheless, stripped serum still significantly induced resumption of development. Thus, there must be something else being sensed and this may be stimulating endogenous production of DA.

2) Endogenous DA synthesis may still be required for resumption of iL3 development (as has clearly been shown in other species). The authors try and argue around this by claiming that DA biosynthetic enzyme mRNA levels are low at the point of infection. No evidence was presented that any of the mRNAs surveyed actually encode a biosynthetic enzyme (or that the protein levels matched the RNA levels). However, giving the authors the benefit of the doubt that these homologous sequence do encode the responsible enzymes, their interpretation of their own data still contradicts their claim. The enzyme BmaDAF-9, which they claim is the DAF-9 ortholog, is very highly expressed at L3i! This is exactly what one might expect in order to jumpstart the immediate synthesis of DA upon infection. Note that the other enzymes in the upstream synthetic pathway are not rate-limiting and thus are not necessarily even needed at this point, as long as the substrate for DAF-9, which is rate-limiting, is present in the parasite (or may be acquired from the host upon infection). It is also noteworthy that as the parasite starts its developmental program, all of these enzymes increased. This also makes sense, since as has been shown in S. stercoralis, DA levels become highest later in parasitic development within the host. Lastly, as the authors also acknowledged, the mRNA levels do not necessarily correspond to protein levels, and this has been observed in other species also.

3) As mentioned in point #1, something in the serum may be stimulating DA synthesis. The authors suggest temperature or some other environmental factor is responsible for stimulating development, which may work without DA. This is a misinterpretation of how these environmental factors are already known to work in other species. Studies in both C. elegans and S. stercoralis have shown that these factors act upstream in the resumption of dauer/parasitic development by turning on DA synthesis (which is still required). In these and other nematode species, DAF-12 and DAF-9 are required for resumption of development, which is also convincing evidence that the ligand is required.

A couple of other points are worth mentioning that the authors use to argue against endogenous ligand synthesis. First, the lack of an effect of dafadine as an inhibitor of endogenous DA synthesis in the parasite is not at all conclusive. No evidence was presented that dafadine inhibits DA synthesis in the parasite. C. elegans DAF-9 is not the same enzyme as BmaDAF-9 or DAF-9 found in other parasites and may not be effective against parasites. To that point, dafadine does not inhibit DAF-9 or endogenous DA synthesis in S. stercoralis, demonstrating the lack of broad inhibitory activity.

Despite these concerns, the paper is of interest if the conclusions and discussion were modified accordingly. A simple suggestion for how the authors might improve the paper to be acceptable (without further experimentation), is to fairly discuss the points above and change their definitive conclusions in the abstract and title. Why not just state “Filarial DAF-12 senses host serum to resume iL3 development” and mention the alternatives that are just as likely, or perhaps work in concert? This changes would still get their points across and would not limit their speculation pending further definitive results.

**Part II – Major Issues: Key Experiments Required for Acceptance**

Reviewer #3: (No Response)

**Part III – Minor Issues: Editorial and Data Presentation Modifications**

Reviewer #3: (No Response)

PLOS authors have the option to publish the peer review history of their article (what does this mean?). If published, this will include your full peer review and any attached files.

Reviewer #3: No

Figure Files:

Data Requirements:

Reproducibility:

References:

---

## [Editor Report · Decision Letter 2]

5 Jun 2023

Dear Dr Betous,

We are pleased to inform you that your manuscript 'Filarial DAF-12 sense the host serum to resume iL3 development during infection' has been provisionally accepted for publication in PLOS Pathogens.

Best regards,

Richard J. Martin, BVSc, PhD, DSc, DipECVPT, FRCVS

Guest Editor

PLOS Pathogens

P'ng Loke

Section Editor

PLOS Pathogens

Kasturi Haldar

Editor-in-Chief

PLOS Pathogens

orcid.org/0000-0001-5065-158X

Michael Malim

Editor-in-Chief

PLOS Pathogens

orcid.org/0000-0002-7699-2064

The concerns of referee 3 have been addressed in the revised discussion.
---

## [Editor Report · Acceptance letter]

14 Jun 2023

Dear Mr Bétous,

We are delighted to inform you that your manuscript, "Filarial DAF-12 sense the host serum to resume iL3 development during infection," has been formally accepted for publication in PLOS Pathogens.

Best regards,

Kasturi Haldar

Editor-in-Chief

PLOS Pathogens

orcid.org/0000-0001-5065-158X

Michael Malim

Editor-in-Chief

PLOS Pathogens

orcid.org/0000-0002-7699-2064